# Association between Reaction Times in the Joint Simon Task and Personality Traits

**DOI:** 10.3390/brainsci13081207

**Published:** 2023-08-15

**Authors:** Shun Irie, Atsumichi Tachibana, Akiko Matsuo

**Affiliations:** 1Division for Smart Healthcare Research, Dokkyo Medical University, Tochigi 321-0293, Japan; 2Department of Anatomy, Dokkyo Medical University, Tochigi 321-0293, Japan; a-tachi@dokkyomed.ac.jp; 3Research Center for Advanced Science and Technology, The University of Tokyo, Tokyo 113-8654, Japan; matsuo@bfp.rcast.u-tokyo.ac.jp

**Keywords:** presence, joint Simon effect, personality, Big Five

## Abstract

Joint go and no-go effects (joint Simon effects; JSEs) are considered to have a stimulus–response compatibility effect on joint reaction time tasks (joint Simon task) caused by the presence of other people. Additionally, JSEs are known to be associated with various social factors and are therefore a potential clinical marker for communicative function; however, the relationship with the personality that is associated with communication skills remains unclear. In this study, we focused on the association between JSE and personality traits. Thirty Japanese participants (fifteen women) were recruited. First, personality trait scores were obtained using the Japanese version of the ten-item personality inventory before the experiment. Second, we measured reaction times in the joint Simon task and single go and no-go tasks with the go signal presented on the congruent and incongruent sides. At last, we analyzed the association between reaction times and personality traits by using Spearman’s correlation analysis. As a result, we observed two pairs with significant correlations: JSE and neuroticism and short reaction times in the joint condition and agreeableness. In conclusion, we identified potential psychological markers associated with the joint Simon task. These findings may lead to an additional hypothesis regarding the neurobiological mechanisms of JSEs.

## 1. Introduction

Human interactions, a highly developed behavioral function, are an essential part of daily communication which enable people to overcome difficult problems that cannot be solved alone. However, many people suffer from communication function disorders such as autism spectrum disorder and schizophrenia [1]. Additionally, some people are not skilled at social interactions, even without pathophysiological conditions [2]. Understanding these variabilities in communicative function is important in a diverse society.

In psychological studies, human interactions have been assessed using the joint action paradigm [3]. In psychology, the joint action paradigm is defined as two or more individuals coordinating their behaviors (i.e., response, movement, or language) to achieve shared goals, which can either hinder or promote their actions, as seen in reaction times in joint go and no-go tasks [3]. In this paradigm, verbal communication, cooperation, and/or competition are often present [4,5,6].

Additionally, recent studies have suggested that the synchronization of brain activity (oscillations) plays an important role in human interactions [6]. Therefore, the simultaneous recording of psychological and physiological assessments of multiple people, such as hyperscanning of electroencephalography (EEG), functional near-infrared spectroscopy (fNIRS), and functional magnetic resonance imaging (fMRI), is gaining attention as a strong methodology for understanding human interactions [6,7,8]. However, these studies have mainly focused on brain synchrony between individuals, without considering individuals’ variability associated with their characteristics (i.e., personality and episodic factors). Additionally, it remains unclear whether the brain synchrony and behavioral outcomes of the joint action paradigm are influenced by social factors or by low-level mechanisms (not so social) [9,10]. If reliable scales for social factors are associated with the outcomes of the joint action paradigm, they could potentially serve as a useful tool for objectively evaluating psychological states and brain functions related to communication disorders. This study specifically focused on personality traits measured by the Big Five model because it is a widely accepted and reliable psychological marker for personality traits, including sociability, which is important for daily communication in educational, sports, and work settings [11,12,13,14]. Previous research has already identified brain areas and neurophysiological features in EEG that are associated with the Big Five model [15,16], and these might also be related to various psychiatric diseases and symptoms [17,18]. Based on these findings, we hypothesized that individual personality traits would be associated with the outcomes obtained from the joint action paradigm experiments, providing a new objective assessment of social functions.

This study investigated the joint Simon effect (JSE), an outcome of the joint action paradigm that serves as a psychological marker for social emotions in both biological and non-biological objects placed adjacent to individual participants [19]. Generally, the JSE is assessed through joint go and no-go tasks [19,20]. In this task, two participants sitting in front of a monitor responded to go signals presented on both the right and left sides of the center. In the case of a participant sitting on the right side, the reaction times (RTs) to the go signal appearing on the left side (incongruent side) tended to be delayed compared to those appearing on the right side (congruent side). However, this was predominantly observed when the left person was present, which was considered to be similar to the stimulus–response (S–R) compatibilities in the Simon task [19,20]. Interestingly, several reports have described that the delay in RTs varied according to social contexts, such as empathy for partners, anthropomorphism, and other factors [5,21,22,23,24]. Thus, we expected that the modulation of RTs induced by the existence of a partner would be modulated by personal and social factors related to communicative functions.

For personal social factors, we focused on the Big Five personality traits: extraversion (E), agreeableness (A), conscientiousness (C), neuroticism (N), and openness (O) [25]. Specifically, extraversion and agreeableness are important factors closely related to communication functions [26,27]. From our hypothesis described above, the modulation of RTs measured by the joint go and no-go tasks (JSE) should be related to the personality traits of individual participants, which might lead to an understanding of both psychological and neural mechanisms. Thus, we examined these relationships using a joint go and no-go task and a questionnaire on personality traits.

## 2. Materials and Methods

### 2.1. Participants

This study included 30 Japanese participants (15 women and 15 men, aged 18–49 years) with no history of neuromuscular disorders. All participants except two were right-side dominant, in accordance with the Edinburgh inventory [28]. Written informed consent was obtained before the experiments. In accordance with the Declaration of Helsinki, the study protocol was approved by the Research Ethics Committee of Dokkyo Medical University (approval No. 2021-023).

During the experiments, the participants were asked to sit on a chair placed on the right side facing a table (w × d × h: 118 × 74 × 70 cm) in a quiet room and place their index fingers on a reaction switch (Chronos, Psychology Software Tools, Inc., Pittsburgh, PA, USA) (Figure 1A). A liquid crystal display (w × h: 52.5 × 29.5 cm; SA230, Acer, Taiwan) was placed in the center of the table.

### 2.2. Joint Go and No-Go Task

We designed a joint go and no-go task (a joint Simon task) according to a previous study [10]. In the single condition, only one participant sat on the chair and performed the go and no-go task (left panel of Figure 1A). In the joint condition, the participant performed the task in the same manner; however, the experimenter joined the task (right panel of Figure 1A) and responded during the no-go trials using the left side of the response switch.

In the joint and single conditions of go/no-go tasks, the task sequences consisted of wait (−1.5–0 s to warning signal (WS; 1000 Hz, 0.2 s)), go/no-go (0.7 s after WS), and feedback (1.5 s after WS) phases (Figure 1B). For the wait phase, the participants were asked to wait with the center of the fixation cross (2.5 × 2.5 cm) presented at the center of the screen. For the go/no-go phase, the response signals were presented 7.5 cm to the right and left sides of the screen. The response signals consisted of squares (*l* = 2.5 cm) and circles (*d* = 2.5 cm). Participants were asked to respond only when the response signal of the square was presented, but not when they responded to the circular object in both experimental conditions. However, the researcher responded to the response signal using a circular object. In the feedback phase, feedback messages that meant correct or incorrect (OK or NG, respectively) were automatically presented on the display.

Four response types were defined for data analysis. When the square objects for the go signal were presented on the right side of the display, we defined them as go/congruent trials. However, when square objects were presented on the left side, we defined them as go/incongruent. Lastly, we defined the trials in which the circular objects were displayed on the left and right sides as no-go/congruent and no-go/incongruent, respectively (Figure 1C).

The experimental sequences were repeated 256 times, with 64 responses per response type per condition. To prevent mental fatigue, we subdivided the task into four sessions (64 responses per session) with 1 minute intervals between sessions. The order of the response types was randomized. In addition, the order of the experimental conditions was counterbalanced. Randomization and RT measurements were conducted using Chronos with the E-Prime 3.0 system (Psychology Software Tools, PA, USA).

### 2.3. Psychological Assessment

The Japanese version of the Ten-Item Personality Inventory (TIPI-J) [29], comprising the Big Five personality dimensions, was used for psychological assessment. Each trait measure consists of two items and participants rated each item on a scale of 1 (strongly disagree) to 7 (strongly agree). The total score for each trait was calculated as described in a previous study [29,30,31]. The assessment was conducted before joint go and no-go tasks.

### 2.4. Data Analysis

For the RT analysis, we removed trials with incorrect (2.42%; 93/3840) and late (>1 s) responses. We then calculated the mean and standard deviation (SDs) of the RTs for each condition and response type. Following a previous study, we excluded the responses over and below the mean (±2.5 SD) and the wrong responses for each condition and response type. Of the responses, 4.66% (179/3840) were excluded from further analysis. From the remaining reactions, we calculated the mean RT for each condition and response type. Additionally, we calculated the congruent effects (incongruent–congruent) in both the single and joint conditions and the joint effects (joint–single) in both congruent and incongruent trials. Finally, we calculated the differences in congruent effects across conditions (joint–single).

### 2.5. Statistics

To compare the RTs across experimental conditions and response types, we performed a repeated measures (RM) ANOVA with Holm’s post hoc test (paired *t*-tests with Holm’s correction). Congruent and joint effects on RTs were then compared across conditions by performing paired *t*-tests. Additionally, we calculated Spearman’s correlation coefficient (rho) with an estimation of 95% confidence intervals (CI) using bootstrap estimations (resample: 20, repetition: 2000) for assessing the relationship between the congruent and joint effects and delta congruent effects with the score of each personality trait [32].

The group data are expressed as the means ± SE unless otherwise noted. The level of significance was set at *p* < 0.05. Statistical analyses were performed using R version 4.1.0 (R Core Team 2021. R: Language and Environment for Statistical Computing. R Foundation for Statistical Computing, Vienna, Austria).

## 3. Results

Figure 2 shows the means of the RTs in each condition (A), the distributions of the scores obtained on the TIPI-J (B), and the congruency (C) and joint (D) effects. For RTs, the RM-ANOVA indicated significant main effects in both conditions, response types, and interactions between factors (congruence, *F*_1,28_ = 23.25, *p* < 0.001; condition, *F*_1,28_ = 17.96, *p* < 0.001; congruency × condition, *F*_1,28_ = 10.35, *p* = 0.003). Additionally, we observed significant differences between the congruent and incongruent trials in both the joint and single conditions (Holm post hoc tests, *p* < 0.01). As shown in Figure 2B, the JSE was higher in the joint condition than in the single condition (*t*_29_ = 3.255, *p* = 0.003). Moreover, the joint effects in the incongruent trials were significantly larger than those in the congruent trials (*t*_29_ = 3.255, *p* = 0.003).

Table 1 shows the Spearman’s correlation coefficients between the factors, 95% CI, and *P*-values. For congruent effects, the correlation coefficients between neuroticism and congruent effects in both conditions were significant (*p* < 0.05). Additionally, agreeableness and joint effects in both trials reached significance (*p* < 0.05). Furthermore, congruent (Delta in Table 1) effects correlated with conscientiousness at a marginally significant level (*p* < 0.1).

## 4. Discussion

In this study, we observed two significant effects on the RTs in single and joint go and no-go tasks. The first effect, known as the congruent effect, refers to the RT delay when the go signal appeared on the experimenter’s left side. This delay was more pronounced when the experimenter sat next to a participant. The other effect was the joint effect, defined as shorter RTs when the experimenter sat next to a participant (Figure 2). Interestingly, we also observed a significant correlation between these effects on RTs and personal traits. The pairing of congruent effects with neuroticism reached a moderate level (*rho* = −0.279, −0.361 in joint and single conditions, respectively). Similarly, pairing joint effects with agreeableness also reached a moderate level (*rho* = 0.314, 0.402 in congruent and incongruent conditions, respectively) (Table 1). Additionally, there was no significant correlation observed in other combinations; however, conscientiousness was slightly correlated at a marginally significant level.

### 4.1. Possible Mechanisms of Joint and Congruency Effects on RTs

This study had three important findings. First, the congruency effects on the RT tasks in the joint condition were compared to those in the single condition, similar to previous studies on the joint Simon task [19,20,33]. Second, RTs shortened in the joint condition in both congruent and incongruent trials, but the magnitude of the shortening of RTs was significantly larger in the congruent trials than in the incongruent trials (JSE). Finally, the congruent and joint effects on RTs were significantly correlated with the neuroticism and agreeableness personality traits.

Sebanz (2003) first reported the congruent effect in the joint condition [19]; RTs for an incongruent side were delayed only when the participants performed the RT task with a neighboring partner, which is called the “joint Simon effect” and considered one of the psychological markers for the presence of others. The mechanisms of the delayed response on the incongruent side are well explained as S–R compatibilities, similar to the normal Simon effect [19,33,34,35], wherein the participant is asked to respond to two types of stimuli (go signals for the right and left fingers). When the stimulus presentation was shifted to the right or left side, the RTs for the opposite direction to the stimulus presentation were delayed; this is called the Simon effect [35]. In the JSE, two participants divided the stimulus response instead of choosing their fingers, which also led to a delayed response in the opposite stimulus directions [19,33]. In both effects, an inconsistency between the location of the stimulus presentation and the response hand induces a conflict between the participants, which is considered the major cause of delayed responses [19,33]. In this study, we observed that the congruency effects (JSE) increased in the joint condition, indicating that we could replicate previous JSE studies. Additionally, this study revealed a correlation between congruent effects and neuroticism, as measured by the TIPI-J (Table 1).

According to the Big Five model, neuroticism is an essential personality trait related to feelings of negative emotions (i.e., anxiety, worry, and anger) [36,37]. Several studies have suggested that neuroticism scores are associated with various psychopathological disorders, such as social anxiety and depression [38,39]. Furthermore, neuroticism is positively associated with activity in the medial prefrontal cortex (mPFC) [40]. Interestingly, the mPFC was also negatively associated with congruent effects in the joint condition in a voxel-based morphometry study [41]. In this study, the congruent effects in both conditions were negatively correlated with neuroticism scores (Table 1). Therefore, congruent effects should be modulated by a common brain region between neuroticism and congruent effects, such as the mPFC.

However, these results also suggest that congruent effects were observed in the single condition. The congruent effect in the joint go and no-go task was considered to be due to S–R compatibility in the spatial domains as well as the normal Simon task (response hand and cue locations) [19,33]. Neuroticism is negatively correlated with spatial attention function in older individuals [42]. This correlation may be due to the spatial domains of cognitive functions associated with neuroticism.

Another novel study finding is the foreshortening of RTs in the joint condition compared to those in the single condition (Figure 2A). Factors foreshortening RTs in an individual included task complexity, arousal, learning effect, fatigue, and attention [43,44,45,46,47,48]. Regarding task complexity, there were no differences in how the participants performed in either experimental condition. Thus, task complexity was not considered relevant to the current findings. Additionally, the possibility that learning effects and fatigue were related to these results could also be excluded because the order of the experimental conditions (joint or single) was counterbalanced. In contrast, the shortening of the RT was modulated through arousal and attention, dependent on the existence of a partner. The existence of partners is known to promote cognitive processes [49]. Several studies have indicated the shortness of RTs in the joint condition; however, statistical analyses directly comparing the RTs between the joint and single conditions were not conducted [5,20,21]. In addition, foreshortening was significantly correlated with agreeableness, that is, being altruistic and sympathetic to others, which is beneficial for teamwork and customer service. This was described in terms of sociability and morality [26,27,50,51]. Interestingly, agreeableness was associated with brain metrics in the dorsolateral prefrontal cortex (DLPFC), temporoparietal junction, and lingual gyrus, which are different from the brain metric properties related to neuroticism [15]. Thus, we concluded that the congruent and joint effects on RTs in the joint go and no-go tasks were differently modulated in an independent neural manner. Additionally, conscientiousness, described as the tendency to be responsible for others and hardworking [52], was slightly correlated with the difference in congruent effects between the joint and single conditions at a marginally significant level (*p* < 0.1). Importantly, it is also a personality trait associated with the morphometric properties of the DLPFC [15]. Thus, further studies are needed to assess the relationship between the congruent and joint effects in this task and neural properties, such as regional volumes and functional connectivity measured by fMRI, EEG, and fNIRS [7,53,54].

### 4.2. Limitations

This study had two limitations. First, we could not find psychological markers associated with congruent effects in joint conditions. Personality traits are often expressed as combinations or clusters of multiple traits [55]; a study on a mass population detected four types of clusters of personality traits [55]. This study revealed that the personality traits of neuroticism and agreeableness were negatively and positively correlated with the congruent and joint effects, respectively, in both conditions (Table 1). For example, agreeableness and extraversion are communication-related traits sometimes expressed in four dimensions: (1) high extraversion/high agreeableness, (2) high extraversion/low agreeableness, (3) low extraversion/high agreeableness, and (4) low extraversion/low agreeableness [51]. Additionally, the characteristics and populations of personality traits differ by nationality [56]. We collected data only from Japanese participants. Thus, it is necessary to investigate these clusters and their dimensions in the global mass population. Furthermore, this study did not include an adequate sample size for the clustering analysis because the measurement of the joint go and no-go tasks took too long to gather data from a larger sample. Thus, further studies are needed to construct brief measurement methods for joint go and no-go tasks to gather data from a larger sample.

Second, the TIPI-J may have had less sensitivity to detecting the relationship between personality traits and the congruent and joint effects in the joint go and no-go tasks in small samples because it is only one of the brief scaling methods [25]. Thus, further studies focusing on specific personality traits using precise scaling methods, such as the Big Five Inventory (BFI) and Revised NEO Personality Inventory (NEO-PI-R), are needed [57,58].

### 4.3. Implications

Although there were methodological limitations, our results suggest that the outcomes of the joint and single go and no-go tasks could potentially be psychological markers that objectively explain the individual variabilities of communication functions and/or social skills among healthy individuals and those with communication disorders. In a previous study, the congruent effects in the joint condition were significantly associated with the volume of mPFC, which has a relationship with social anxiety [41,59,60]. Recent research has also suggested that social anxiety is positively correlated with neuroticism and negatively correlated with other domains, including agreeableness [61]. This study demonstrated relationships between joint and congruent effects with the Big Five model personality traits. However, future studies are needed to reveal the complex relationship among the outcomes of joint go and no-go tasks, personality traits, neural activities, and the symptoms of several social disorders. These studies will provide a more in-depth understanding of social functions applicable to the objective evaluation and treatment of social function symptoms.

## 5. Conclusions

This study demonstrated a correlation between congruent and joint effects on RTs in a joint go and no-go task and neuroticism and agreeableness, respectively. These results suggest common neural mechanisms between both effects and personality traits, leading to a better understanding of various developmental and/or psychiatric disorders. Further studies are needed to determine the neural mechanisms underlying these associations.

## Figures and Tables

**Figure 1 brainsci-13-01207-f001:**
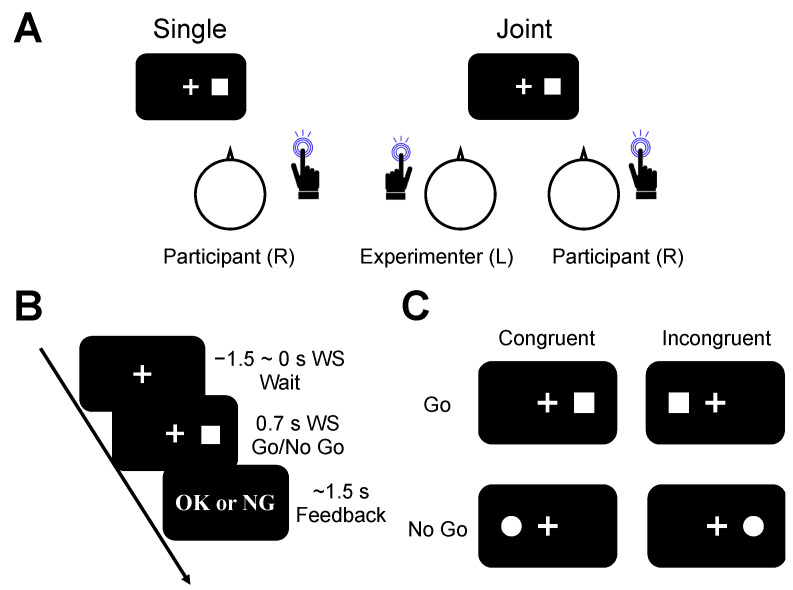
Methodologies for the joint go and no-go task. (**A**) Experimental conditions. Participants sat on the right and were directed to the liquid crystal display without (single) and with an experimenter sitting on the left (joint). The participant and experimenter were asked to fix the cross at the center of the screen. (**B**) The task sequences consisted of three phases: wait (−1.5–0 s from the warning signal (WS)), go/no-go (0.7 s after WS), and feedback (1.5 s after WS). (**C**) Reaction cues were divided into two categories: go (white square) and no-go (white circle). Additionally, these cues were subdivided depending on the stimulus location of the fixation cross: right (congruent) or left (incongruent).

**Figure 2 brainsci-13-01207-f002:**
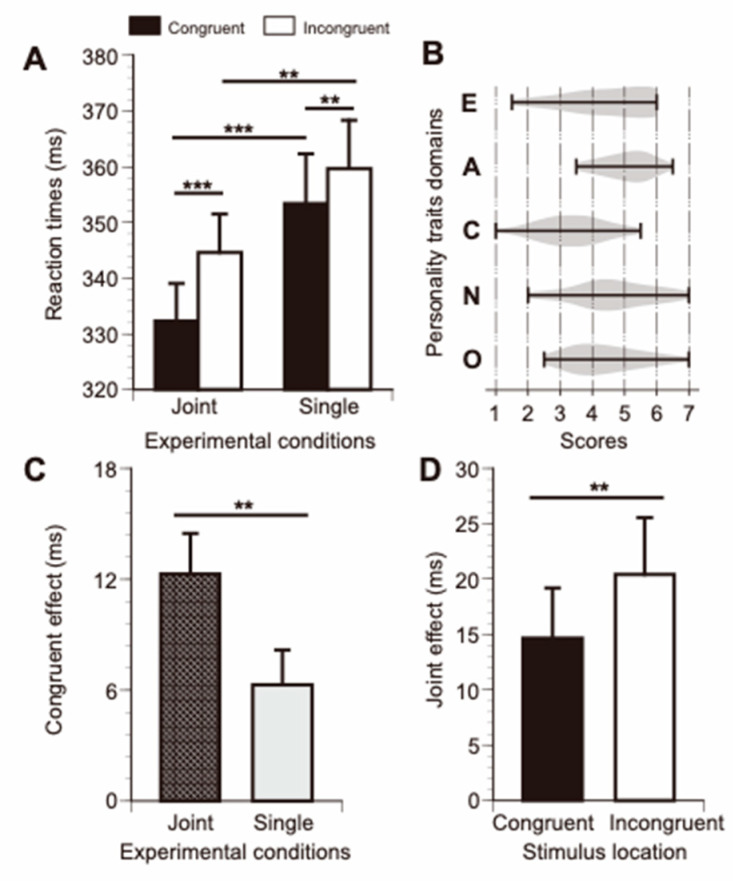
Reaction times and personality trait analysis. (**A**) Average reaction times (RTs) for each experiment and trial; black and white filled bars represent average RTs in congruent and incongruent trials, respectively. (**B**) Violin plots showing the distribution of personality trait scores measured using the TIPI-J. (**C**,**D**) Congruent and joint effects. ** *p* < 0.01, *** *p* < 0.001.

**Table 1 brainsci-13-01207-t001:** Spearman’s correlation coefficients between factors.

	Congruency Effect			Joint Effect	
	Joint	Single	Delta (J–S)	Congruent	Incongruent
E	0.061 (−0.229–0.350)	−0.003 (−0.306–0.301)	0.102 (−0.167–0.372)	−0.069 (−0.369–0.231)	−0.083 (−0.375–0.208)
A	0.047 (−0.225–0.318)	0.087 (−0.167–0.341)	−0.040 (−0.326–0.247)	0.314 * (0.072–0.557)	0.402 *** (0.203–0.602)
C	0.026 (−0.249–0.301)	−0.088 (−0.361–0.185)	0.218 ^†^ (−0.029–0.464)	0.104 (−0.155–0.363)	0.072 (−0.172–0.315)
N	−0.279 * (−0.503–−0.055)	−0.361 ** (−0.591–−0.131)	−0.024 (−0.304–0.255)	0.142 (−0.112–0.395)	0.179 (−0.081–0.438)
O	0.044 (−0.243–0.332)	0.033 (−0.251–0.316)	0.049 (−0.242–0.341)	−0.091 (−0.355–0.172)	−0.151 (−0.425–0.122)

^†^ *p* < 0.1, * *p* < 0.05, ** *p* < 0.01, *** *p* < 0.001; E: extraversion; A: agreeableness; C: conscientiousness; N: neuroticism; O: openness.

## Data Availability

The dataset is available upon request to the corresponding author.

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
