# Peer review of "Association between Reaction Times in the Joint Simon Task and Personality Traits"

_brainsci, 2023, doi:10.3390/brainsci13081207_

Round 1
Reviewer 1 Report
Dear Authors
In the abstract, the terms "joint Simon effect" and "Big Five" were not found in the Mesh database.
In the introduction, To make it easier for readers who may not be familiar with the term, it would be helpful to first provide a brief definition or explanation of joint-action paradigms before discussing their use in psychological studies. Additionally, the research question or hypothesis could benefit from more explicit signposting in the introduction. While the study's objectives are mentioned, it is not immediately clear what specific question the study seeks to answer. The introduction lays a solid foundation for the study, but a few small adjustments could improve its clarity and readability. it would be helpful to rephrase the research question to make it more explicit. Currently, the introduction outlines the study's objectives and hypotheses but does not clearly state the precise research question. Clarifying the research question would help to focus the study and make it easier for readers to understand the purpose of the research.
The discussion provides a clear and concise summary of the study's findings. However, there are a few areas where the discussion could be improved to enhance its clarity and completeness. Firstly, it would be helpful to provide a brief explanation of what congruent and joint effects are, and how they were measured in the joint go and no-go task. This would ensure that readers who are not familiar with these terms understand the study's methodology and results. Additionally, it would be helpful to provide more specific details about the correlations between personality traits and congruent and joint effects. For example, which personality traits were most strongly correlated with each effect, and what were the effect sizes of these correlations? This would help readers understand the relationships between personality traits and joint action performance in more detail. Finally, the discussion could benefit from a more in-depth interpretation of the study's findings and their implications. For example, what are the theoretical and practical implications of the finding that different personality traits are associated with congruent and joint effects? How might these findings be used to inform future research or interventions aimed at improving joint action performance? In summary, while the discussion provides a clear summary of the study's findings, there is room for improvement in terms of providing more detail and interpretation of the results.
Author Response
Thank you for a polite review and meaningful suggestions. There were opportunities to improve readability, especially concerning the hypothesis and implications. We have provided point-by-point responses to your comments.
[Abstract]
â™1: In the abstract, the terms "joint Simon effect" and "Big Five" were not found in the Mesh database.
Response 1: Thank you for pointing this out. As you mentioned, joint Simon effect and Big Five were not included in the MeSH database; however, both terms are important for seeking similar research. Of course, we can rephrase these to joint go and no-go task and personal traits, respectively. Thus, we would like the reviewers and editors to consider our rephrasing.
[Introduction]
â™2: To make it easier for readers who may not be familiar with the term, it would be helpful to first provide a brief definition or explanation of joint-action paradigms before discussing their use in psychological studies.
Response 2: Thank you for pointing this out. We have inserted sentences to explain the joint action paradigm (Page 1, lines 35-39).
â™3: Additionally, the research question or hypothesis could benefit from more explicit signposting in the introduction.
Response 3: Thank you for pointing this out. Reviewer 2 also commented on this. In this study, we hypothesized that individual personality traits would be associated with the outcomes, which would yield a novel objective evaluation for social functions. Thus, we have fixed the phrases and added information to support the hypothesis. Please refer to Page 1-2, lines 45-61.
â™4: While the study's objectives are mentioned, it is not immediately clear what specific question the study seeks to answer. Currently, the introduction outlines the study's objectives and hypotheses but does not clearly state the precise research question. Clarifying the research question would help to focus the study and make it easier for readers to understand the purpose of the research.
Response 4: As described in Response 3, the hypothesis was unclear. Thus, we have fixed the applicable sentences to improve the readability and elaborate upon the study’s background. Please refer to Page 2, lines 45-61.
[Material & Methods]
Nothing.
[Results]
Nothing
[Discussion]
â™5: Firstly, it would be helpful to provide a brief explanation of what congruent and joint effects are, and how they were measured in the joint go and no-go task. This would ensure that readers who are not familiar with these terms understand the study's methodology and results.
Response 5: As you mentioned, the original manuscript lacked a brief explanation of the results and terms. Thus, we have added further explanation on Page 5, lines 192-203.
â™6: Additionally, it would be helpful to provide more specific details about the correlations between personality traits and congruent and joint effects. For example, which personality traits were most strongly correlated with each effect, and what were the effect sizes of these correlations? This would help readers understand the relationships between personality traits and joint action performance in more detail.
Response 6: As you suggested, we have added additional information in the results section to highlight the correlations as well as the RTs. Please refer to Page 5, lines 197-203.
â™7: Finally, the discussion could benefit from a more in-depth interpretation of the study's findings and their implications. For example, what are the theoretical and practical implications of the finding that different personality traits are associated with congruent and joint effects? How might these findings be used to inform future research or interventions aimed at improving joint action performance?
Response 7: As you mentioned, the original manuscript lacked a full interpretation of the implications. We studied this paradigm to understand the psychological and neural mechanisms for communication functions and disorders. Thus, we have added an implication section (4.3.). Please refer to Page 7, lines 295-309.
Reviewer 2 Report
The MS is overall well-written and structured and the topic interesting and new. Hence I am positive about publication pending the revisions listed below
Abstract. Please shorten it by following a more streamlined way and focusing mostly on the main results.
Introduction. Please refer more to recent papers and explain deeply why it is important studying the associations with personality traits
2.3 should read 'Personality traits (or assessment)'. Was this measure took before or after the Simon task? Please explain the choice
Author Response
Thank you for a polite review and meaningful suggestions. According to your suggestions, I was able to reinforce the manuscript’s readability and logical construction. We have provided point-by-point responses to your suggestions.
[Abstract]
#1: Please shorten it by following a more streamlined way and focusing mostly on the main results.
Response 1: Thank you for pointing out this. As you suggested, we totally reconstructed the body of abstract so that it will be easy access to main results. Please refer to Page 1, lines 14-24.
[Introduction]
#2: Please refer more to recent papers and explain deeply why it is important studying the associations with personality traits
Response 2: Thank you for pointing this out. As you suggested, we have added background information to establish the research questions (Page 1, lines 35-39 and Page 2, lines 45-61).
[Material & Methods]
#3: should read 'Personality traits (or assessment)'. Was this measure took before or after the Simon task? Please explain the choice
Response 3: Thank you for pointing this out. TIPI-J was conducted before measuring the reaction times. Thus, we have added this detail to the applicable paragraph (Page 4, line 141).